# Sample complexity of Schrödinger potential estimation

## Abstract

We address the problem of Schrödinger potential estimation, which plays a crucial role in modern generative modelling approaches based on Schrodinger bridges and stochastic optimal control for SDEs. Given a simple prior diffusion process, these methods search for a path between two given distributions $\rho_0$ and $\rho_T$ requiring minimal efforts. The optimal drift in this case can be expressed through a Schrödinger potential. In the present paper, we study generalization ability of an empirical Kullback-Leibler (KL) risk minimizer over a class of admissible log-potentials aimed at fitting the marginal distribution at time $T$. Under reasonable assumptions on the target distribution $\rho_T$ and the prior process, we derive a non-asymptotic high-probability upper bound on the KL-divergence between $\rho_T$ and the terminal density corresponding to the estimated log-potential. In particular, we show that the excess KL-risk may decrease as fast as $\mathcal{O}(\log n/n)$ when the sample size $n$ tends to infinity even if both $\rho_0$ and $\rho_T$ have unbounded supports.

## 1 Introduction

The Schrödinger Bridge problem (SBP) originates from a question posed by Erwin Schrödinger in 1932 [Schrödinger, 1932], seeking the most likely evolution of a probability distribution between two given endpoint distributions while minimizing relative entropy with respect to a prior stochastic process. This problem has deep connections with optimal transport [Leonard, 2014] and stochastic control [Dai Pra, 1991]. In its simplest continuous-time form, one aims to construct a so-called *Schrödinger Markov process* whose joint begin-end distribution $\pi(\mathrm{d}x, \mathrm{d}z)$ has the representation

$$\pi(\mathrm{d}x, \mathrm{d}z) = \mathsf{Q}(z, T \mid x, 0)\,\nu_0(\mathrm{d}x)\,\nu_T(\mathrm{d}z), \tag{1}$$

where $\mathsf{Q}(z, T \mid x, 0)$ is the transition kernel of a reference Markov process, and $\nu_0, \nu_T$ are unknown "boundary potentials" to be determined. The desired marginals $\pi(\mathrm{d}x, \mathbb{R}^d)$ and $\pi(\mathbb{R}^d, \mathrm{d}z)$ are given, and one seeks $\nu_0$ and $\nu_T$ that reproduce these marginals. In the rest of the paper, we assume that both $\pi(\mathrm{d}x, \mathbb{R}^d)$ and $\pi(\mathbb{R}^d, \mathrm{d}z)$ are absolutely continuous with respect to the Lebesgue measure and denote the corresponding densities by $\rho_0$ and $\rho_T^*$, respectively. Classical existence proofs for the SBP date back to Fortet [1940] (in 1D) and Beurling [1960], with a modern fixed-point approach in [Chen et al., 2016]. Recent extensions to the case of noncompactly supported marginal distributions can be found in [Conforti et al., 2024] and [Eckstein, 2025]. Recently, the problem attracted attention of machine learners in the context of generative modelling (see, for instance, [Tzen and Raginsky, 2019, De Bortoli et al., 2021, Shi et al., 2023, Korotin et al., 2024, Gushchin et al., 2024a, Rapakoulias et al., 2024] to name a few). It follows from Theorem 3.2 in [Dai Pra, 1991] that the optimal Markov process $X_t^*$ solving the Schrödinger problem with marginals $(\rho_0, \rho_T^*)$ can be constructed as a solution of the following SDE:

$$\mathrm{d}X_t^* = \big(b(X_t^*, t) + \sigma(X_t^*, t)\sigma(X_t^*, t)^\top \nabla \log h(X_t^*, t)\big)\,\mathrm{d}t + \sigma(X_t^*, t)\,\mathrm{d}W_t, \quad X_0 \sim \rho_0,$$

where

$$h(w,t) = \int_{\mathbb{R}^d} \mathsf{Q}(y,T \mid w,t)\, \nu_T(\mathrm{d}y)$$

and $\mathsf{Q}$ is the transition density of the reference (or base) diffusion process

$$\mathrm{d}X_t = b(X_t,t)\,\mathrm{d}t + \sigma(X_t,t)\,\mathrm{d}W_t, \quad X_0 \sim \rho_0.$$

The transition density $\mathsf{Q}^*$ of the reciprocal process $X_t^*$ can be obtained from $\mathsf{Q}$ via the so-called Doob's $h$–transform:

$$\mathsf{Q}^*(y,T \mid x,t) = \mathsf{Q}(y,T \mid x,t)\, \frac{h(y,T)}{h(x,t)}. \tag{2}$$

This is precisely the law of the base process conditioned by the function $h$ (see [Jamison, 1974]). In many presentations of the Schrödinger Bridge problem, one takes a very simple reference process (for instance, a Brownian motion) so that its transition kernel is straightforward to write down (see, for example, [Pooladian and Niles-Weed, 2024] and [Baptista et al., 2024]). However, there are several practical and theoretical advantages to considering more general (potentially higher-dimensional, or with domain constraints, or with a non-trivial drift/diffusion) reference processes.

In the present paper, we are interested in estimation of the Schrödinger potential $\nu_T$ from $n$ i.i.d. samples $Y_1, \ldots, Y_n \sim \rho_T^*$. Given a class of log-potentials $\Psi$, we study generalization ability of an empirical risk minimizer

$$\widehat{\psi} \in \operatorname*{argmin}_{\psi \in \Psi} \left\{ -\frac{1}{n} \sum_{i=1}^{n} \log \left( \int_{\mathbb{R}^d} \mathsf{Q}(Y_i, T \mid x, 0)\, \frac{h_\psi(Y_i, T)}{h_\psi(x, 0)}\, \rho_0(x)\mathrm{d}x \right) \right\}, \tag{3}$$

where

$$h_\psi(x,t) = \int \mathsf{Q}(y,T \mid x,t)\, e^{\psi(y)}\,\mathrm{d}y.$$

Let us note that, in view of (2),

$$\rho_T^\psi(y) = \int_{\mathbb{R}^d} \mathsf{Q}(y,T \mid x,0)\, \frac{h_\psi(y,T)}{h_\psi(x,0)}\, \rho_0(x)\mathrm{d}x$$

is the marginal endpoint probability density of a diffusion process $X_t^\psi$ corresponding to Doob's $h_\psi$-transform:

$$\mathrm{d}X_t^\psi = \left( b(X_t^\psi, t) + \sigma(X_t^\psi, t)\sigma(X_t^\psi, t)^\top \nabla \log h_\psi(X_t^\psi, t) \right)\mathrm{d}t + \sigma(X_t^\psi, t)\,\mathrm{d}W_t, \quad X_0 \sim \rho_0.$$

In other words, the estimate $\widehat{\psi}$ minimizes empirical Kullback-Leibler (KL) divergence between the actual target $\rho_T^*$ and the marginal densities $\rho_T^\psi$ over the class of admissible log-potentials $\Psi$. That is, we chose the log-potential $\psi$ that makes the transformed reference diffusion hit the observed terminal law, and measure error only through KL of the marginals. Because $h_\psi$ is used inside the Doob factor, the learnt potential is compatible with a single Markov process; one never risks obtaining mutually inconsistent forward/backward potentials. The method combines the full problem (the marginals, transition densities, and the potential function) into one single optimization framework. By doing so, it aims to directly minimize the objective of matching the marginals at time $T$ without separating the problem into smaller subproblems. In contrast, the Sinkhorn algorithm, commonly used for optimal transport problems, approaches the problem by iteratively updating the potentials in a decoupled manner. At each iteration, a simpler least squares problem appears, which is linear in one potential function given that another one is fixed from the previous iteration. The Sinkhorn algorithm alternates between updating the potential functions to match the marginals of the distributions and adjusting the transport plan until convergence. We refer to Pooladian and Niles-Weed [2024], Chiarini et al. [2024] for recent results. The primary advantage of the Sinkhorn approach is its computational efficiency. By decoupling the optimization process into simpler, linear problems, the Sinkhorn method can handle large-scale problems effectively. This iterative procedure allows for faster updates, and it has become a popular method for many optimal transport applications, see Genevay et al. [2018], March and Henry-Labordere [2023] However, the approach presented in this paper differs in that it does not separate the problem into independent steps. Instead, it aims at solving the Schrödinger system

approximately by formulating it as a single optimization problem involving Doob $h$-transform of the base process $X$ parametrized by the Schrödinger potential. Unlike iterative proportional fitting (Sinkhorn), everything is learnt in one go, avoiding slow or unstable fixed-point cycles. This results in a more accurate and robust solution. The trade-off between the two methods lies in computational efficiency versus the quality of the solution. The Sinkhorn approach provides a quick and efficient solution by solving simpler problems at each iteration, but it may not achieve the best possible solution for the full problem. On the other hand, the method presented in this paper offers a more holistic approach, which could lead to a more accurate matching of the marginal distributions but might require more computational resources.

The approach presented in this paper can also be compared to methods that rely on optimization over transport maps, see Korotin et al. [2024], Gushchin et al. [2024a]. In transport map-based approaches, the goal is to find a map $\mathcal{T}$ that transports one probability distribution to another. The optimization typically focuses on minimizing a quadratic cost functional that penalizes the difference between the target distribution and the transformed distribution under the transport map. These methods are often framed as optimal transport problems, where the map $\mathcal{T}$ is determined by solving an optimization problem that involves the marginal distributions. The advantage of optimization over transport maps lies in its clear geometric interpretation, where the transport map provides a direct way to relate the two distributions. This can lead to efficient algorithms, especially when the transport map can be parametrized in a way that allows for fast computations, such as in the case of certain neural network architectures or simple affine transformations, Rapakoulias et al. [2024].

However, transport map-based approaches are typically constrained to quadratic costs, which may limit their applicability in some cases. Specifically, quadratic cost functionals, such as the 2-Wasserstein distance, often assume a certain structure or symmetry that may not be ideal for more general or complex problems.

In contrast, the approach discussed in this paper is not limited to quadratic costs. It allows for more general cost structures and is based on minimizing the Kullback-Leibler divergence (KL-divergence), which can accommodate a wider range of problem types. This flexibility is particularly valuable when dealing with more complex distributions or when the underlying problem involves non-quadratic costs that capture other aspects of the distribution, such as entropy regularization or non-linear interactions between variables.

**Contribution**    The main contribution of the present paper a sharper non-asymptotic high-probability upper bound on generalization error of the empirical risk minimizer $\widehat{\psi}$ defined in (3).

- Taking a multivariate Ornstein-Uhlenbeck process as the reference one, we show that (see Theorem 1), with probability at least $(1 - 2\delta)$, the excess KL-risk of the marginal endpoint density $\widehat{\rho}_T$ corresponding to $\widehat{\psi}$ satisfies the inequality

$$\mathsf{KL}(\rho_T^*, \widehat{\rho}_T) - \inf_{\psi \in \Psi} \mathsf{KL}(\rho_T^*, \rho_T^\psi) \lesssim \sqrt{\Upsilon(n, \delta) \inf_{\psi \in \Psi} \mathsf{KL}(\rho_T^*, \rho_T^\psi)} + \Upsilon(n, \delta),$$

  where

$$\Upsilon(n, \delta) \lesssim \frac{\log^2 n + \log(1/\delta) \log n}{n}.$$

  Here and further in the paper, the sign $\lesssim$ stands for an inequality up to a multiplicative constant. The derived upper bound has several advantages over the existing results. First, in contrast to Korotin et al. [2024], the excess risk may decrease as fast as $\mathcal{O}(\log^2 n / n)$ provided that the class of log-potentials $\Psi$ is rich enough to approximate the target density $\rho_T^*$. Second, unlike theoretical guarantees for Sinkhorn-based approaches (see e.g. Pooladian and Niles-Weed [2024]), we are able to relate the endpoint marginal densities $\rho_T^*$ and $\widehat{\rho}_T$.

- We impose very mild assumptions on the target density $\rho_T^*$. We only require $\rho_T^*$ to be bounded and sub-Gaussian. On the other hand, the available convergence proofs for the Sinkhorn algorithm rely on the stronger assumption that the marginals are log-concave, see Conforti et al. [2024]. We also avoid the so-called strong density assumptions like boundedness from below often used in nonparametric statistics in the context of log-density estimation.

- The assumptions on the class of log-potentials $\Psi$ are also reasonable. We support our claim with several examples.

**Paper structure**  The rest of the paper is organized as follows. Section 2 is devoted to a short review of related work. In Section 3, we introduce necessary definitions and notations. After that, we present our main result (Theorem 1) in Section 4 and discuss main ideas of its proof in Section 5. Rigorous derivations as well as auxiliary technical results are deferred to the supplementary material.

## 2  Related work

Here is a short review of methods used in the literature to compute Schrödinger potentials, including the Sinkhorn algorithm. The Schrödinger potential, which arises in optimal transport problems, represents a key component in the solution of transport problems involving marginal distributions. Over time, several methods have been proposed to compute these potentials efficiently, with applications in areas ranging from statistical mechanics to machine learning. Here, we review some of the most prominent methods used in the literature.

**Sinkhorn algorithm**  The Sinkhorn algorithm Sinkhorn [1967] is one of the most widely used methods for computing Schrödinger potentials in the context of optimal transport. It is based on iterative scaling and aims to solve the optimal transport problem by alternating between updating two potentials $\nu_0$ and $\nu_T$ to enforce marginal constraints. The key advantage of the Sinkhorn approach is its computational efficiency, particularly when the transport problem is framed with a quadratic cost (such as the 2-Wasserstein distance), see Pavon et al. [2021], Chen et al. [2021], Stromme [2023] for reference. In each iteration, the algorithm solves a simpler problem that involves scaling the potentials in a way that brings the marginals of the transformed distribution closer to the target. Although Sinkhorn's algorithm is efficient and widely applicable, it is often limited by its assumption of quadratic costs. Additionally, the algorithm does not directly handle more complex cost structures, such as non-quadratic costs or non-linear dynamics, which can be a limitation in some applications.

**Sinkhorn bridge**  The Sinkhorn Bridge proposed by Pooladian and Niles-Weed [2024], provides a way to estimate the Schrödinger bridge using Sinkhorn's algorithm in an efficient manner. The key insight of this method is that the potentials obtained from the static entropic optimal transport problem can be modified to yield a natural plug-in estimator for the drift function that defines the Schrödinger bridge. However, this work does not provide bounds on the distance between marginal distributions at time $T = 1$ because there is an exploding term $(1 - \tau)^{k+2}$ as $\tau \to 1$ where $k$ is the dimension of the underlying manifold. This term leads to a "curse of dimensionality" where the error grows rapidly as $\tau$ approaches 1, especially in high-dimensional settings. As a result, the estimation error increases significantly when attempting to estimate the Schrödinger bridge at the terminal time, making it difficult to obtain precise bounds for $T = 1$.

**Dual Formulation of the Schrödinger Problem**  In the dual formulation of the Schrödinger problem, the Schrödinger potential is computed by solving a convex optimization problem. This approach reformulates the problem in terms of a dual objective that involves the Kullback-Leibler (KL) divergence between the target and predicted distributions. The dual problem is then solved using optimization techniques such as gradient descent or variational methods, see Zhang and Chen [2022], Tzen and Raginsky [2019] for reference. This formulation is more flexible than the Sinkhorn algorithm, as it can accommodate more general cost functions and is not limited to quadratic losses.

While the dual approach is flexible, it is often computationally more demanding than Sinkhorn's method due to the need for iterative optimization over high-dimensional spaces. This makes the dual formulation suitable for smaller or more specialized problems, but it can become computationally expensive in large-scale applications.

**Approximate Solutions Using Monte Carlo Methods**  Monte Carlo methods, particularly those relying on reverse diffusion processes, have also been employed to approximate Schrödinger potentials. In these methods, a reverse process is simulated, and the potential is iteratively refined to minimize the discrepancy between the predicted and target marginals, see Korotin et al. [2024] for reference. These methods are often used when the problem involves complex dynamics that are difficult to capture using direct optimization techniques.

Monte Carlo methods are particularly useful when dealing with high-dimensional problems, as they allow for the sampling of large spaces. However, they can be computationally expensive and may require a significant number of samples to achieve an accurate solution.

In addition, there are approaches that rely heavily on Monte Carlo approximations of intermediate values rather than the Schrödinger potentials themselves, among which the following should be noted De Bortoli et al. [2021], Vargas et al. [2021], Peluchetti [2023].

**Neural Network-Based Approaches**   Recent advancements in deep learning have led to the use of neural networks to approximate Schrödinger potentials. These approaches treat the potential function as a parameterized neural network and use gradient-based optimization techniques to learn the potential that best matches the marginals. The use of neural networks offers a flexible and powerful way to model complex non-linear potentials, making these methods well-suited for problems with intricate dynamics or non-quadratic costs. While neural network-based approaches are highly flexible, they require large amounts of data and computational resources to train the network, and they are often prone to overfitting if not regularized appropriately. Despite these challenges, they represent a promising direction for future research, especially when the problem at hand involves complex and high-dimensional systems. We refer to Liu et al. [2023], Wang et al. [2021] for recent results.

**Iterative Markovian Fitting**   The Iterative Markovian Fitting (IMF) method, introduced in the recent work by Shi et al. [2023], offers an approach to solving Schrödinger Bridge (SB) problems. Unlike previous methods, such as Iterative Proportional Fitting (IPF), IMF guarantees the preservation of both the initial and terminal distributions in each iteration, which is a key advantage over IPF where these marginals are not always preserved. IMF alternates between two types of projections: Markovian projections and reciprocal projections, ensuring that the resulting distribution remains within the correct class (Markovian or reciprocal) while progressively approximating the Schrödinger Bridge. We refer to Gushchin et al. [2024b] for recent results.

In Silveri et al. [2024], the authors provide the convergence analysis for diffusion flow matching (DFM), a method used to generate approximate samples from a target distribution by bridging it with a base distribution through diffusion dynamics. Their theoretical work includes non-asymptotic bounds on the Kullback-Leibler (KL) divergence between the true target distribution and the distribution generated by the DFM model. A key insight from this paper is the incorporation of two sources of error: drift-estimation and time-discretization errors. However, while the convergence analysis offers theoretical guarantees, the statistical error is not explicitly addressed in this paper. The analysis assumes that all expectations are exact, which might not hold in practical settings where samples are finite, and statistical errors could arise due to the approximations involved in the generative process. Thus, future work will need to extend this analysis to quantify the impact of statistical approximations in finite-sample settings.

## 3   Preliminaries and notations

This section collects necessary definitions and notations. As we announced in the contribution paragraph, we are going to consider a multivariate Ornstein-Uhlenbeck process as a reference one. For this reason, we elaborate on its basic properties in this section.

**Multivariate Ornstein-Uhlenbeck process**   To be more specific, we will consider the base process $X_t^0$ solving the SDE

$$dX_t^0 = b\left(m - X_t^0\right) dt + \Sigma^{1/2}dW_t, \quad 0 \leqslant t \leqslant T,$$

where $b > 0$ controls the drift rate, $m \in \mathbb{R}^d$ represents the mean-reversion level, $\Sigma \in \mathbb{R}^{d \times d}$ is a positive definite symmetric matrix, and $W_t$ is a standard $d$-dimensional Wiener process. It is known that the conditional distribution of $X_t^0$ given $X_0^0 = x$ is Gaussian $\mathcal{N}\left(m_t(x), \Sigma_t\right)$ with

$$m_t(x) = (1 - e^{-bt})m + e^{-bt}x \quad \text{and} \quad \Sigma_t = \frac{1 - e^{-2bt}}{2b}\Sigma. \tag{4}$$

This implies that the corresponding Doob's $h$-transform can be expressed through the Ornstein-Uhlenbeck operator

$$\mathcal{T}_t g(x) = \frac{1}{(2\pi)^{d/2}\sqrt{\det(\Sigma_t)}} \int_{\mathbb{R}^d} \exp\left\{-\frac{1}{2}\|\Sigma_t^{-1/2}(y - m_t(x))\|^2\right\} g(y)\,\mathrm{d}y.$$

Indeed, it holds that $h_\psi(x, t) = \mathcal{T}_{T-t}e^{\psi(x)}$. Then, introducing

$$\mathsf{q}(y\,|\,x) = \frac{1}{(2\pi)^{d/2}\sqrt{\det(\Sigma_T)}} \exp\left\{-\frac{1}{2}\|\Sigma_T^{-1/2}(y - m_T(x))\|^2\right\},$$

we note that

$$\rho_T^\psi(y) = \int_{\mathbb{R}^d} \frac{\mathsf{q}(y\,|\,x)e^{\psi(y)}}{\mathcal{T}_T e^{\psi(x)}}\,\rho_0(x)\,\mathrm{d}x \tag{5}$$

is the marginal density of $X_T^\psi$, the endpoint of a random process $X_t^\psi$ governed by $h_\psi$:

$$\mathrm{d}X_t^\psi = b\left(m - X_t^\psi\right)\mathrm{d}t + \nabla\log\left(\mathcal{T}_{T-t}e^{\psi(X_t^\psi)}\right)\mathrm{d}t + \Sigma^{1/2}\mathrm{d}W_t, \quad X_0^\psi \sim \rho_0.$$

If the Schrödinger potential $\nu_T$ admits a density $e^{\psi^*}$ with respect to the Lebesgue measure, then the optimally controlled process $X_t^*$ solves the SDE

$$\mathrm{d}X_t^* = b\left(m - X_t^*\right)\mathrm{d}t + \nabla\log\left(\mathcal{T}_{T-t}e^{\psi^*(X_t^*)}\right)\mathrm{d}t + \Sigma^{1/2}\mathrm{d}W_t, \quad X_0^* \sim \rho_0.$$

Finally, it is well known that the unique stationary (invariant) distribution of $X_t^0$ is Gaussian, that is, $X_t^0$ converges to $X_\infty^0$ in distribution as $t \to \infty$ with $X_\infty \sim \mathcal{N}(m, \Sigma/(2b))$. Since the parameters of the limiting distribution do not depend on the starting point, $\mathcal{T}_\infty g(x) \equiv \mathcal{T}_\infty g$ is a constant.

**Other notations**  The notation $f \lesssim g$ or $g \gtrsim f$ means that $f = \mathcal{O}(g)$. Besides, we often replace $\max\{a, b\}$ and $\min\{a, b\}$ by shorter expressions $a \vee b$ and $a \wedge b$, respectively. For any $s \geqslant 1$, the Orlicz $\psi_s$-norm of a random variable $\xi$ is defined as

$$\|\xi\|_{\psi_s} = \inf\left\{u > 0 : \mathbb{E}e^{|\xi|^s/u^s} \leqslant 2\right\}.$$

Finally, given $p \geqslant 1$ and a probability density $\rho$, the weighted $L_p$-norm of a function $f$ is defined as $\|f\|_{L_p(\rho)} = \left(\mathbb{E}_{\xi\sim\rho}|f(\xi)|^p\right)^{1/p}$. Given two probability densities $\rho_0 \ll \rho_1$ on $\mathbb{R}^d$, the Kullback-Leibler divergence between them is defined as $\mathsf{KL}(\rho_0, \rho_1) = \mathbb{E}_{\xi\sim\rho_0}\log\left(\rho_0(\xi)/\rho_1(\xi)\right)$.

## 4  Main result

In the present section, we discuss statistical properties of the empirical risk minimizer $\widehat{\psi}$ defined in (3). In particular, Theorem 1 provides a Bernstein-type upper bound on its excess KL-risk. We impose the following assumptions. First, as we announced before, we use the Ornstein-Uhlenbeck process $X_t^0$ as the reference one.

**Assumption 1.** *The base process $X^0$ solves the following SDE*

$$\mathrm{d}X_t^0 = b\left(m - X_t^0\right)\mathrm{d}t + \Sigma^{1/2}\,\mathrm{d}W_t, \quad 0 \leqslant t \leqslant T.$$

*where $b > 0$, $m \in \mathbb{R}^d$, $\Sigma$ is a positive definite symmetric matrix of size $d \times d$, and $W$ is a $d$-dimensional Brownian motion.*

Main properties of the Ornstein-Uhlenbeck process were discussed in the previous section. Second, we suppose that the target density $\rho_T^*$ meets the following requirements.

**Assumption 2.** *The target distribution at time $T$ admits a bounded density $\rho_T^*$ with respect to the Lebesgue measure such that*

$$\rho_T^*(x) \leqslant \rho_{\max} \quad \text{for all } x \in \mathbb{R}^d.$$

*Moreover, the target distribution $\rho_T^*$ is sub-Gaussian with variance proxy $\mathrm{v}^2$, that is,*

$$\mathbb{E}_{Y\sim\rho_T^*}e^{u^\top Y} \leqslant e^{\mathrm{v}^2\|u\|^2/2} \quad \text{for any } u \in \mathbb{R}^d. \tag{6}$$

Assumption 2 is very mild. Despite the fact that we deal with logarithmic loss, we do not require $\rho_T^*$ to be bounded away from zero. We do not even require its support to be compact. This significantly complicates the proof of the excess KL-bound and poses nontrivial technical challenges. Let us note that the condition 6 yields that $\mathbb{E}_{Y \sim \rho_T^*} Y = 0$. However, it does not diminish generality of our setup.

The remaining assumptions concern properties of the class of log-potentials $\Psi$. First, we assume that admissible log-potentials $\psi(x)$ are bounded from above and behave as $\mathcal{O}(\|x\|^2)$ as $x$ tends to infinity.

**Assumption 3.** *There exist non-negative constants $\Lambda$ and $M$ such that*

$$-\Lambda \left\| \Sigma^{-1/2}(x-m) \right\|^2 - M \leqslant \psi(x) \leqslant M \quad \text{for all } x \in \mathbb{R}^d \text{ and } \psi \in \Psi.$$

*Moreover, for any $\psi \in \Psi$, it holds that $\mathcal{T}_\infty \psi = \mathbb{E}\psi(X_\infty) = 0$.*

The condition $\mathcal{T}_\infty \psi = 0$ appears because of the fact that the Schrödinger potentials $\nu_0$ and $\nu_T$ (see (1)) are defined up to a multiplicative constant. The requirement $\mathcal{T}_\infty \psi = 0$ is nothing but a normalization. Second, we assume that $\Psi$ is parametrized by a finite-dimensional parameter $\theta \in \mathbb{R}^D$:

$$\Psi = \{\psi_\theta : \theta \in \Theta\},$$

where $\Theta$ is a subset of a $D$-dimensional cube $[-R, R]^D$ and each function $\psi_\theta$ maps $\mathbb{R}^d$ onto $\mathbb{R}$. We suppose that the parametrization is sufficiently smooth in the following sense.

**Assumption 4.** *There exists $L \geqslant 0$ such that*

$$|\psi_\theta(x) - \psi_{\theta'}(x)| \leqslant L \left( 1 + \|x\|^2 \right) \|\theta - \theta'\|_\infty \quad \text{for all } \theta, \theta' \in \Theta \text{ and all } x \in \mathbb{R}^d.$$

Assumptions 3 and 4 are quite general. We provide two examples when they hold. First, in a recent paper [Korotin et al., 2024], the authors model $e^{\psi(x)}$ as a Gaussian mixture. Let $\alpha_1, \ldots, \alpha_K$ be non-negative numbers such that $\alpha_1 + \ldots + \alpha_K = 1$ and consider

$$e^{\psi(x)} = e^{-C} \sum_{k=1}^{K} \alpha_k \varphi_{m_k, \Sigma_k}(x), \quad \text{where} \quad \varphi_{m_k, \Sigma_k}(x) = \frac{e^{-\|\Sigma_k^{-1/2}(x-m_k)\|^2/2}}{(2\pi)^{d/2} \det(\Sigma_k)^{1/2}}.$$

Here $C$ is a normalizing constant which ensures that $\mathcal{T}_\infty \psi = 0$. In this situation, the parameter $\theta$ consists of all $\alpha_k$'s and all components of $m_k$'s and $\Sigma_k$'s, $k \in \{1, \ldots, K\}$. If the smallest eigenvalues of $\Sigma_1, \ldots, \Sigma_K$ are bounded away from zero uniformly over $k \in \{1, \ldots, K\}$, then $e^{\psi(x)}$ is bounded. On the other hand, if $K$ is fixed, there is a component with a weight at least $1/K$. Without loss of generality, we assume that it is the first one. Then

$$\psi(x) \geqslant -C + \log\left(\alpha_1 \varphi_{m_1, \Sigma_1}(x)\right) \geqslant -C - \log K - \frac{1}{2} \left\| \Sigma_1^{-1/2}(x - m_1) \right\|^2,$$

and we conclude that Assumption 3 is satisfied. Verification of the Assumption 4 is straightforward once we assume that the weight of each component is bounded away from zero, and the norms $\|m_k\|$, $\|\Sigma_k\|$, and $\|\Sigma_k^{-1}\|$ are bounded uniformly over $k \in \{1, \ldots, K\}$ (which is the case in [Korotin et al., 2024]). Second, Assumptions 3 and 4 will be fulfilled if one deals, for example, with a class of truncated feedforward neural networks with bounded weights and ReLU activations. It is known that (see [Schmidt-Hieber, 2020, Lemma 5]) they are Lipschitz with respect to each weight, and the Lipschitz constant grows linearly with $\|x\|$. More generally, Conforti [2024] analyzed semiconvexity properties of the Schrödinger potentials under rather mild assumptions on the marginals.

We are ready to formulate the main result of this section.

**Theorem 1.** *Let $\rho_0$ be the density of the standard Gaussian distribution $\mathcal{N}(0, I_d)$. Grant Assumptions 1, 2, 3, and 4. Assume that $T$ is sufficiently large in a sense that*

$$bT \geqslant (5 + \log d) \vee \log\left(160b\left(\mathrm{v}^2 \vee 1\right) \|\Sigma^{-1}\|\right).$$

*Let $\widehat{\psi}$ be defined in (3) and let $\widehat{\rho}_T$ be the corresponding density of $X_T^{\widehat{\psi}}$. Then, for any $\delta \in (0, 1/2)$, with probability at least $1 - 2\delta$, it holds that*

$$\mathsf{KL}(\rho_T^*, \widehat{\rho}_T) - \inf_{\psi \in \Psi} \mathsf{KL}(\rho_T^*, \rho_T^\psi) \lesssim \sqrt{\Upsilon(n, \delta) \inf_{\psi \in \Psi} \mathsf{KL}(\rho_T^*, \rho_T^\psi)} + \Upsilon(n, \delta),$$

*where*

$$\Upsilon(n, \delta) = (\Lambda d + M + d) \left( d + \log \frac{RLn}{\delta} + (M \vee \log \Lambda) \sqrt{d} e^{-bT} \right) \frac{D \log n}{n}.$$

*The hidden constant behind $\lesssim$ depends on $\Sigma$, $m$, $b$, and $\mathrm{v}$ only.*

In Theorem 1, we assume that $\rho_0$ is the density of $\mathcal{N}(0, I_d)$. Though it is a standard choice of initial distribution in practice, we would like to emphasize that unbounded support of $\rho_0$ significantly complicates the proof and makes the problem even more challenging.

The problem of Schrödinger potential estimation was also studied in [Korotin et al., 2024] and [Pooladian and Niles-Weed, 2024]. In [Korotin et al., 2024], the authors suggest an algorithm called Light Schrödinger Bridge, which is based on minimization of the empirical KL-divergence between entropic optimal transport plans. This slightly differs from our setup, since we aim to minimize empirical KL-divergence between marginal endpoint distributions. The reason is that Korotin, Gushchin, and Burnaev [2024] are motivated by the style transfer task, where the initial distribution is also unknown. In contrast, we focus on generative modelling where the initial distribution $\rho_0$ is available to learner. In [Korotin et al., 2024, Theorem A.1], the authors consider the case when admissible potentials are Gaussian mixtures with $K$ components. Assuming that both initial and finite distibutions have a compact support, they prove a $\mathcal{O}(n^{-1/2})$ upper bound on the Rademacher complexity of such class. On the other hand, we allow the support of $\rho_0$ and $\rho_T^*$ to be unbounded. Besides, the rate of convergence presented in Theorem 1 may be much faster than $\mathcal{O}(n^{-1/2})$ if the target distribution is close to $\{\rho_T^\psi : \psi \in \Psi\}$. In the realizable case (that is, $\rho_T^* \in \{\rho_T^\psi : \psi \in \Psi\}$) the right-hand side in Theorem 1 becomes $\mathcal{O}(\log^2 n / n)$. Finally Theorem 1 provides a high-probability upper bound on the excess risk while the result of [Korotin et al., 2024] holds in expectation. In [Pooladian and Niles-Weed, 2024] the authors study properties of a plug-in Sinkhorn-based estimator. Similarly to Korotin et al. [2024], they consider the case of compactly supported initial and target measures. However, they assume that these measures are supported on smooth $k$-dimensional submanifolds. They derive a $\mathcal{O}(n^{-1/2} + (T - \tau)^{-k-2} n^{-1})$ bound on the squared total variation distance between *path measures* up to moment $\tau < T$. Unfortunately, the second term grows very fast when $\tau$ approaches $T$, and there are no guarantees whether the marginal endpoint distributions will be close to each other.

In Theorem 1, we focus on the statistical error leaving study of the approximation out of the scope of the present paper. The reason is that there are few results on properties of the true log-potential $\psi^*(x) = \log\left(\nu_T(\mathrm{d}x)/\mathrm{d}x\right)$. However, we would like to note that, according to our findings (see Lemma B.2 and (5)), if $\psi^*$ fulfils Assumption 3, then for any $\psi \in \Psi$ and $y \in \mathbb{R}^d$

$$\log \frac{\rho_T^*(y)}{\rho_T^\psi(y)} \lesssim |\psi(y) - \psi^*(y)|$$
$$+ (\mathcal{T}_\infty |\psi - \psi^*|)^{1/\mathcal{K}(T)} \left\| \Sigma^{-1/2}(y - m) \right\|^{2 - 2/\mathcal{K}(T)} e^{\mathcal{O}(e^{-bT} \|\Sigma^{-1/2}(y-m)\|^2)},$$

where $1 \leqslant \mathcal{K}(T) \leqslant 1 + \mathcal{O}(\sqrt{d} e^{-bT})$. In the proof of Theorem 1 (see Step 4), we show that the expectation

$$\mathbb{E}_{Y \sim \rho_T^*} \left\| \Sigma^{-1/2}(Y - m) \right\|^{2 - 2/\mathcal{K}(T)} e^{\mathcal{O}(e^{-bT} \|\Sigma^{-1/2}(Y-m)\|^2)}$$

is finite, provided that $bT \geqslant (5 + \log d) \vee \log\left(160 b\left(\mathrm{v}^2 \vee 1\right) \left\|\Sigma^{-1}\right\|\right)$. This allows us to relate the KL-divergence between $\rho_T^*$ and $\rho_T^\psi$ with the distances between the corresponding log-potentials:

$$\mathsf{KL}\left(\rho_T^*, \rho_T^\psi\right) \lesssim \|\psi - \psi^*\|_{L_1(\rho_T^*)} + (\mathcal{T}_\infty |\psi - \psi^*|)^{1/\mathcal{K}(T)}.$$

# 5 Proof sketch of Theorem 1

In this section, we discuss main ideas used in the proof of Theorem 1. Rigorous derivations are deferred to Appendix A. Since the proof is quite long, we split it into several steps.

**Step 1: log-density properties.** Let us note that Assumptions 3 and 4 concern properties of log-potentials $\psi \in \Psi$ while empirical risks include marginal densities $\rho_T^\psi$. For this reason, before we consider the empirical process

$$\frac{1}{n} \sum_{i=1}^n \log \frac{\rho_T^*(Y_i)}{\rho_T^\psi(Y_i)} - \mathsf{KL}\left(\rho_T^*, \rho_T^\psi\right), \quad \psi \in \Psi,$$

we have to study the random variables $\log\left(\rho_T^*(Y_i)/\rho_T^\psi(Y_i)\right)$, $1 \leqslant i \leqslant n$. Using basic properties of the Ornstein-Uhlenbeck operator, we show that

$$- \log \rho_T^\psi(y) \lesssim -\psi(y) + \left\| \Sigma^{-1/2}(y - m) \right\|^2.$$

In view of Assumption 3, this means that $-\log \rho_T^\psi(y)$ grows as fast as a quadratic function. Since the target distribution is sub-Gaussian and has a bounded density, this yields that the random variables $\log\left(\rho_T^*(Y_i)/\rho_T^\psi(Y_i)\right)$, $1 \leqslant i \leqslant n$, are sub-exponential. More specifically, applying Lemma C.3 we obtain the following upper bound on their Orlicz norm:

$$\left\| \log \frac{\rho_T^*(Y_i)}{\rho_T^\psi(Y_i)} \right\|_{\psi_1} \lesssim \Lambda d + M + d \quad \text{for all } i \in \{1, \ldots, n\}.$$

**Step 2: $\varepsilon$-net argument and Bernstein's inequality.** The result obtained on the first step allows us to use concentration inequalities for sub-exponential random variables. Let us fix $\varepsilon \in (0, R)$ and let $\Theta_\varepsilon$ stand for the minimal $\varepsilon$-net of $\Theta$ with respect to the $\ell_\infty$-norm. We denote the set of corresponding log-potentials by $\Psi_\varepsilon$:

$$\Psi_\varepsilon = \{\psi_\theta : \theta \in \Theta_\varepsilon\}.$$

Using Bernstein's inequality for unbounded random variables (see, for instance, [Lecué and Mitchell, 2012, Proposition 5.2]) and the union bound, we obtain that

$$\left| \mathsf{KL}\left(\rho_T^*, \rho_T^\psi\right) - \frac{1}{n}\sum_{i=1}^{n} \log \frac{\rho_T^*(Y_i)}{\rho_T^\psi(Y_i)} \right| \lesssim \sqrt{\mathrm{Var}\left(\log \frac{\rho_T^*(Y_1)}{\rho_T^\psi(Y_1)}\right) \frac{\log(2|\Psi_\varepsilon|/\delta)}{n}} + \frac{(\Lambda d + M + d)\log n \log(2|\Psi_\varepsilon|/\delta)}{n}$$

with probability at least $(1 - \delta)$ simultaneously for all $\psi \in \Psi_\varepsilon$.

**Step 3: bounding the loss variance.** One of the key ingredients in the proof of Theorem 1, which allows us to hope for faster rates of convergence than $\mathcal{O}(n^{-1/2})$, is analysis of the variance of $\log\left(\rho_T^*(Y_1)/\rho_T^\psi(Y_1)\right)$, $\psi \in \Psi$. Despite the fact that the admissible log-potentials may be unbounded, we are still able to show that the class $\Psi$ satisfies a Bernstein-type condition

$$\mathrm{Var}\left(\log \frac{\rho_T^*(Y_1)}{\rho_T^\psi(Y_1)}\right) \lesssim (\Lambda d + M + d)\log n \left(\mathsf{KL}\left(\rho_T^*, \rho_T^\psi\right) + \frac{1}{n}\right).$$

**Steps 4 and 5: from $\varepsilon$-net to a uniform Bernstein-type bound.** The hardest and technically involved part of the proof is to show that the losses $\log\left(\rho_T^*(y)/\rho_T^\psi(y)\right)$ and $\log\left(\rho_T^*(y)/\rho_T^\phi(y)\right)$ do not differ too much, once the corresponding log-potentials $\psi$ and $\phi$ are close to each other. This follows from Lemma B.2, which relies on properties of the Ornstein-Uhlenbeck operator established in and Lemma B.3. We would like to note that the unbounded support of the initial density $\rho_0$ significantly complicates the proof of Lemma B.2. Nevertheless, we prove that

$$\log \frac{\rho_T^\psi(y)}{\rho_T^\phi(y)} \lesssim |\psi(y) - \phi(y)| + (\mathcal{T}_\infty|\psi - \phi|)^{1/\mathcal{K}(T)} \|\Sigma^{-1/2}(y-m)\|^{2-2/\mathcal{K}(T)} e^{\mathcal{O}(e^{-bT}\|\Sigma^{-1/2}(y-m)\|^2)},$$

where $1 \leqslant \mathcal{K}(T) \leqslant 1 + \mathcal{O}(\sqrt{d}e^{-bT})$. Though the right-hand side depends exponentially on the squared norm of $\Sigma^{-1/2}(y - m)$, the coefficient $\mathcal{O}(e^{-bT})$ is quite small, which is enough for our purposes.

**Steps 6 and 7: choice of $\varepsilon$ and the final bound.** The rest of the proof is quite standard. On Step 6, we choose an appropriate $\varepsilon$ and obtain a uniform Berstein-type inequality

$$\mathsf{KL}\left(\rho_T^*, \rho_T^\psi\right) - \frac{1}{n}\sum_{i=1}^{n} \log \frac{\rho_T^*(Y_i)}{\rho_T^\psi(Y_i)} \lesssim \sqrt{\Upsilon(n, \delta)\, \mathsf{KL}\left(\rho_T^*, \rho_T^\psi\right)} + \Upsilon(n, \delta),$$

where

$$\Upsilon(n, \delta) = (\Lambda d + M + d)\left(d + \log \frac{RLn}{\delta} + (M \vee \log \Lambda)\sqrt{d}e^{-bT}\right)\frac{D\log n}{n},$$

which holds simultaneously for all $\psi \in \Psi$ with probability at least $(1 - 2\delta)$. After that, we transform it into the desired excess risk bound and finish the proof.

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
