# OpenReview forum: "Sample complexity of Schrödinger potential estimation"
_NeurIPS.cc/2025/Conference — Submitted to NeurIPS 2025_

### Official Review · Reviewer_LmKh · 2025-06-17

**Clarity:** 2
**Significance:** 3
**Originality:** 3
**Rating:** 5
**Confidence:** 4

**Summary:**

The paper proposes studies the Schr\"odinger problem. That between two given marginal distributions $\rho_0$ and $\rho_T$. Given a reference path measure, taken here to be a multivariate Ornstein Uhlenbeck process, the  task is to find its best approximation in relative entropy sense among all path measures having marginal $\rho_0$ at time $0$ and $\rho_T$ at time $T$. The optimal solution is known to be characterized by the fact that the logarithm of its Radon-Nykodym density against the reference is the sum of two functions depending respectively only on the initial and final positions. These functions are known as Schr\"odinger potentials. As a consequence, the optimal process is a Doob h-transform of the reference measure. The paper proposes a method to estimate Schr\"odinger potentials starting from $n$ samples of $\rho_T$ which consist in minimizing among a class of potentials $\Psi$ the empirical Kullback-Leibler divergence between $\rho_T$ and the law at time $T$ of the Doob h-transform associated with $\psi \in \Psi$. The main contribution of this work is Theorem 1, which provides an upper bound for the difference between the law of the Doob transform associated with potential $\hat{\psi}$ estimated from samples and the law of the Doob h-transform associated with $\bar{\psi} \in \arg\min_{\psi\in\Psi}KL(\rho_T|\rho^\psi_T)$ decays like $1/\sqrt{n}$ up to logarithmic factors in $n$. The assumptions required for this result to hold are (among others)

- upper bound on the density and subgaussianity of $\rho_T$
- That he time horizon is large enough
- That the family $\Psi$ is a parametric familiy such that $\psi_\theta$ is in some sense Lipschitz in $\theta$ and that $\theta$ lives in a compact set.

The paper also contains a sketch of proof showing the relevance of concentration ineuqalities in view of obtaining the main result.

**Questions:**

- Can something be said about the optimality of the convergence rates? Can exact calculations be carried out when $\rho_T$ is Gaussian to better understand this point?

- Can the authors elaborate on what strategies could lead to eliminate the assumption that $T$ is large enough? If marginals are compactly supported would their technique allow for this? If so, how?

- Can one expect the dependence on the dimension $d$ in the bounds to be removed under stronger assumptions?

- Assume that the Schr\"odinger potential $\psi$ is Lipschitz. What can be said about the approximation error?

**Ethical Concerns:**

["NO or VERY MINOR ethics concerns only"]

**Final Justification:**

The authors have addressed the points I had raised about the approximation errors. In particular, they clarified that regularity of Schrödinger potentials can help in removing the large $T$ assumption. I maintain my positive score and recommend acceptance of the article.

**Limitations:**

yes

**Quality:**

3

**Strengths And Weaknesses:**

- The strong points of the paper are a) to propose a natural estimator for Schr\"dinger potentials and b) to obtain (partial) guarantees of convergence under reasonable assumptions. I believe there is potential for further theoretical and practical developments of the ideas put forward in this paper.

- A weak point of the paper is that only the statistical error is analyzed. In particular, the authors do not explain whether or not under the current assumptions on $\Psi$ and $\rho_T$ something can be said about $ KL(\rho_T | \rho_T^{\bar{\psi}}) $ with $\bar{\psi} \in \arg\min_{\psi\in \Psi}KL(\rho_T|\rho_T^{\psi})$. As a result, there is no theoretical guarantee saying that $ \lim_{n \rightarrow +\infty} \hat{\psi}$ is close to the real Schr\"odinger potential.  Because of this, the comparison with Sinkhorn algorithm is not so meaningful. Theoretical guarantees for Sinkhorn allow to bound the distance from the optimal coupling whereas here we cannot do the same with the optimal potential. I agree that many results on Sinkhorn lack of a comprehensive statistical analysis however.

-  The authors say that one of the reasons why they do not go beyond the statistical error is that there are not many known results about the regularity of the Schr\"odinger potentials. I disagree with this. In fact, in some of the cited references, many results of this kind are established. I would happy to see, at least in the log-concave case, if these results can be leveraged to quantify the approximation error.

-  The fact that $T$ has to be large enough is in my opinion important and a limitation. It is not discussed enough in the paper. If the authors do not want to least it an assumption they should at least provide more understanding about why this is needed.

---

> ### Author Rebuttal · Authors · 2025-07-31
>
> ### Weakness 1.
>
> We agree with the reviewer that our current theoretical analysis focuses on the statistical error of the estimated Schrödinger potentials and does not provide a full approximation guarantee with respect to the true underlying potentials. This is indeed a limitation, and we acknowledge that deriving such guarantees remains an important direction for future work. That said, we would like to emphasize key differences between our method and the classical Sinkhorn algorithm, which, in our view, justify the comparison despite the differing theoretical foundations. First, the Sinkhorn algorithm is typically implemented on discretized domains, producing couplings between finite point clouds. While it computes potentials satisfying the Schrödinger system on these grids, it does not yield nonparametric estimates of the Schrödinger potentials in the continuous setting. In contrast, our method is designed to learn smooth log-potentials from data in a nonparametric fashion, which allows us to infer information about the dynamics of the entropic interpolation in continuous space. Second, while Sinkhorn's algorithm solves the Schrödinger system directly via an iterative fixed-point procedure, our approach is formulated as an empirical risk minimization problem. Specifically, we minimize the empirical Kullback-Leibler (KL) divergence between the learned path measure and the prior, over a class of admissible log-potentials, with the goal of matching the observed marginal at time $T$. This formulation enables a statistical learning perspective and provides a natural framework for analyzing generalization from finite data.  Moreover, although theoretical convergence results for Sinkhorn exist - such as those in  the recent work [Conforti et al., 2023] - these are derived under idealized conditions. The analysis assumes exact knowledge of the marginals and cost function, and does not incorporate discretization or sampling errors that arise in practice. Thus, while such results offer important insights into the convergence behavior of Sinkhorn iterations, they do not translate directly into statistical guarantees for potential recovery from data.
>
> ### Weakness 2.
>
> We thank the reviewer for this valuable comment on regularity of the Schrödinger potentials. It is true that several of the cited references- particularly in the optimal transport and Schrödinger problem literature - establish important regularity results for the Schrödinger potentials like semiconcavity or semiconvexity. However, our statement referred specifically to the lack of quantitative bounds on the potentials in common functional norms such as Hölder or Sobolev norms, which are crucial when aiming to control approximation error using expressive function classes such as neural networks. When designing nonparametric estimators with approximation guarantees- e.g., based on neural networks- one typically needs bounds on the Sobolev or Hölder norm of the true potential in order to relate approximation error to estimation error.  We were able to show that, if the marginals are compactly supported and smooth on their supports, then the Hölder $\mathcal{H}^{k,\alpha}([0,1]^d)$-norm of the forward log-potential is controlled by $\mathcal{H}^{k,\alpha}([0,1]^d)$-norms of the target density and the Markov kernel of the reference process. We believe this is an important step toward a more complete understanding of approximation error in Schrödinger bridge estimation. Extending these results to broader settings, such as the log-concave case suggested by the reviewer, is indeed a promising and technically interesting direction for future research, and we appreciate the reviewer's encouragement in this regard.
>
> ### Weakness 3.
>
> We thank the reviewer for highlighting this important point concerning $T$. We agree that the requirement for the time horizon
> $T$ to be sufficiently large is indeed a limitation and deserves more discussion. This condition arises from the structure of the Schrödinger problem: when $T$ is too small, the reference OU process does not have enough time to reach stable regime. That said, we would like to emphasize that-crucially-this bound on  $T$ does not depend on the number of samples used for estimation. Once
> $T$ is large enough to ensure some properties of the underlying OU transition kernel, increasing the number of samples improves statistical accuracy but does not impose further constraints on the time horizon. In this sense, the requirement on $T$ is a structural condition of the model rather than a statistical one. We will discuss the condition on $T$ in the revised version.
>
> ### 4. Can something be said about the optimality of the convergence rates? Can exact calculations be carried out when $\rho_T$ is Gaussian to better understand this point?
>
> We thank the reviewer for the insightful question concerning the optimality of the rates. In our work, we specifically consider the excess risk, defined as the difference between the empirical KL divergence of the learned potential and the KL divergence of the best approximation within the chosen function class. This allows us to isolate and study the statistical error-that is, the error arising purely from finite sampling. Assuming that the true potential lies within the chosen function class (i.e., no approximation error), the best possible rate one can hope for is of order $1/n$, up to logarithmic factors, where $n$ is the number of samples. This corresponds to the minimax optimal rate for estimating a log-density under Kullback-Leibler loss in nonparametric settings. Theorem 1 in our paper shows that, in the absence of approximation error, our estimator indeed achieves a $1/n$ convergence rate in KL loss (up to some logarithmic factors). This rate is therefore optimal for the statistical component of the error and cannot be improved in general. We agree that analyzing the case where the marginals are Gaussian could be helpful for gaining more insight into constants and structure, particularly in settings where the exact Schrödinger potentials are known in closed form. This is an interesting direction that could support further study of both statistical and approximation errors in concrete examples.
>
> ### 5. Can the authors elaborate on what strategies could lead to eliminate the assumption that $T$ is large enough? If marginals are compactly supported would their technique allow for this? If so, how?
>
> We thank the reviewer for raising this important question. The condition on the time horizon $T$ is crucial for Lemma B.2, which ensures that the difference $\log \rho_T^\psi - \log \rho_T^\varphi$ is small whenever the corresponding log-potentials $\psi$ and $\varphi$ are close to each other. In its turn, Lemma B.2 relies on Lemma B.3 claiming useful properties of the logarithm of the Ornstein-Uhlenbeck operator. To eliminate the requirement that $T$ must be sufficiently large, one has to study properties of $\log \mathcal T_t g(x) e^{f(x)}$ when $t$ is small.
>
> We would like to note that, if the marginals are compactly supported, then the technical difficulties related to tail behavior and integrability will be significantly reduced. This will simplify the proof of Lemma B.2 significantly. However, at this moment, we do not see how the assumption about compactly supported marginals can lead to milder assumptions on $T$.
>
> ### 6. Can one expect the dependence on the dimension $d$ in the bounds to be removed under stronger assumptions?
>
> To our knowledge, dimension-free bounds usually require additional assumptions, such as $L_2$-$\psi_2$ equivalence in linear regression, which is hard to satisfy in the nonlinear case. Dimension-dependent rates are also usual for generative diffusion models (see, e.g., [Benton et al., 2024] and [Conforti et al., 2025]). We doubt that one can easily eliminate dependence on the dimension.
>
> ### 7. Assume that the Schrödinger potential $\psi$ is Lipschitz. What can be said about the approximation error?
>
> We thank the reviewer for this important question. Yes, assuming that the Schrödinger potential is Lipschitz continuous, something can indeed be said about the approximation error—at least in the compactly supported case. In particular, when the marginals are supported on a compact domain, we can invoke known results on the approximation of Lipschitz continuous functions by deep neural networks in $L^\infty$-norm. These results imply that the approximation error of the Schrödinger potential can be controlled in terms of the network architecture (e.g., depth and width). This is based on the last inequality of Section 4 of our paper, where we explicitly quantify the KL approximation error in terms of $L^1(\rho_T)$ and $L^1(\mathcal N(m, \Sigma))$ norms.
> In the noncompact setting, the situation is more subtle. Direct $L^\infty$-approximation over the entire space is generally not feasible. In such cases, one typically studies an increasing sequence of compact subsets that exhaust the space, and considers a truncated version of the KL divergence, where the integrals are restricted to these compact sets. Controlling the approximation error then requires balancing the truncation error with the approximation quality on each compact set. This is a more delicate analysis, and we regard it as an important topic for future work.
>
> [Benton et al., 2024] J. Benton, V. D. Bortoli, A. Doucet, and G. Deligiannidis. Nearly d-linear convergence bounds for diffusion models via stochastic localization. In The Twelfth International Conference on Learning Representations, 2024.
>
> [Conforti et al., 2023] G. Conforti, A. Durmus, G. Greco. Quantitative contraction rates for Sinkhorn algorithm: beyond bounded costs and compact marginals. ArXiv:2304.04451, 2023.
>
> [Conforti et al., 2025] G. Conforti, A. Durmus, M. G. Silveri. KL Convergence Guarantees for Score Diffusion Models under Minimal Data Assumptions. SIAM Journal on Mathematics of Data Science 7(1), pp. 86-109, 2025.

---

> ### Comment · Reviewer_LmKh · 2025-08-04
>
> Dear authors.
>
> Thank you for your answers. In many respects, these answers have helped me in better understanding then quality of this work. However, I still believe that the comparison with Sinkhorn's algorithm is somewhat imprecise and not fully justified. Also, if the authors do not want to provide some theoretical result on the approximation error, I believe they should provide some insights on how this can be achieved. I still believe that it is doable under a reasonable set of assumptions and I invite them to better check the latest results on Sinkhorn, see the two articles below for example. In these papers, there are several regularity estimates for Schrödinger potentials, that hold for any T>0, and that would likely yield bounds on the approximation error under reasonable assumptions.
>
> I also have a final question
>
> - Suppose you know Schrödinger potentials are regular, say e.g.Lispchitz or semi concave. Would you then be able to remove the large $ T $ assumption of your main result?
>
>
> Here are the above-mentioned references.
>
> - 1) Chizat, L., Delalande, A., & Vaškevičius, T. (2024). Sharper Exponential Convergence Rates for Sinkhorn's Algorithm in Continuous Settings. arXiv preprint arXiv:2407.01202.
>
>
> - 2) Chiarini, A., Conforti, G., Greco, G., & Tamanini, L. (2024). A semiconcavity approach to stability of entropic plans and exponential convergence of Sinkhorn's algorithm. arXiv preprint arXiv:2412.09235.

---

> > ### Author Response · Authors · 2025-08-06
> >
> > ### Comparison with the Sinkhorn algorithm.
> >
> > We thank the reviewer for their follow-up remarks and for pointing us to the recent works by Chizat et al. (2024) and Chiarini et al. (2024). These are indeed highly relevant contributions that provide sharp regularity estimates for Schrödinger potentials and exponential convergence guarantees for the Sinkhorn algorithm in continuous settings. We would like to emphasize, however, that the results in these works are derived in an idealized setting in which all integrals (conditional expectations) appearing in the Schrödinger system can be computed exactly. In such a setting, the potentials can be updated exactly at each iteration, and previously obtained estimates can be fully leveraged in the next iteration. This is fundamentally different from the statistical setting we study, where one has access only to samples from the marginals. In our framework, the integrals in each iteration must be approximated-typically via Monte Carlo-which introduces additional stochastic error that interacts with the regularity properties of the potentials. Moreover, functional estimates of the potentials must be transferred from one iteration to the next, introducing further approximation error on top of the error due to truncating the iteration procedure. We also note that in the recent work [Belomestny et al., 2025] it is shown that when the integrals in the Schrödinger system are computed from data, the convergence of iterative algorithms can no longer be geometrically fast, even in the compactly supported case. This illustrates that transferring the idealized convergence guarantees to the statistical setting is far from straightforward. We fully agree that approximation error analysis is an important and complementary direction. The regularity results in the cited works, particularly in the compact or log-concave setting, could indeed serve as a starting point for quantifying approximation error in our framework. However, incorporating such results into a statistical analysis requires careful control of how approximation and sampling errors propagate through the iterations.
> >
> >
> > ### Approximation error proof insights.
> >
> > We believe that we may use the following strategy to prove an upper bound on the approximation error. In the end of Section 4, we discuss that (see the lines 307--312) it is enough to approximate the true log-potential $\psi^\star$ with respect to $L_1(\rho_T^\star)$- and $L_1(\mathcal N(m, \Sigma / (2b)))$-distance to achieve a small approximation error with respect to the KL-loss. In order to do so, we need a prior knowledge on smoothness of $\psi^\star(y)$ and its behaviour when $y \rightarrow \infty$. For instance, assume that for any $R> 0$ the function $\psi^\star(y)$ is Lipschitz on $[-R, R]^d$ with a constant $L(R)$ and that $|\psi^\star(y)| \leq A (\|y\|^\alpha + 1)$ for all $y \in \mathbb R^d$. Then we can take a sufficiently large $R$, approximate $\psi^\star(y)$ on the cube $[-R, R]^d$ using standard results on properties of, say, ReLU neural networks (see, for instance, [Yarotsky, 2017]), and bound the $L_1$-errors outside $[-R, R]^d$ using the fact that $\psi^\star(y)$ grows polynomially as $y$ tends to infinity.
> >
> > [Belomestny, Schoenmakers, 2025] D. Belomestny, J. Schoenmakers. Forward Reverse Kernel Regression for the Schrödinger bridge problem. Preprint. ArXiv:2507.00640, 2025.
> >
> > [Yarotsky, 2017] D. Yarotsky. Error bounds for approximations with deep ReLU networks. Neural networks 94, 103-114, 2017.

---

> > > ### Author Response · Authors · 2025-08-06
> > > **On omitting the large T condition under regularity assumptions**
> > >
> > > In our paper, the large $T$ condition becomes crucial on the fourth step of the proof of Theorem 1 when we aiming to show that the expectation of $\log \rho_T^{\psi_1}(Y) - \rho_T^{\psi_0}(Y)$, where $Y \sim \rho_T^*$, is small, provided that the log-potentials $\psi_1$ and $\psi_0$ are close enough. It can be removed if we assume that the log-potentials from $\Psi$ are $L$-Lipschitz on $\mathbb R^d$ (with respect to the Euclidean norm) and the initial density $\rho_0$ has a compact support. We elaborate on the proof sketch in the next paragraph.
> > >
> > > Let us fix arbitrary $\psi_0, \psi_1 \in \Psi$ and $y \in \mathbb R^d$ and consider the function $F(s) = \log \rho_T^{\psi_s}(y)$, where $\psi_s(y) = s \psi_1(y) + (1 - s) \psi_0(y)$. Thus, it is enough to show that the derivative of $F(s)$ is bounded.
> > > Direct calculations show that the absolute value of $F'(s)$ does not exceed the absolute value of $\psi_1(y) - \psi_0(y)$ plus the supremum of the absolute value of $\mathcal T_T (\psi_1(x) - \psi_0(x)) e^{\psi_s(x)} / \mathcal T_T e^{\psi_s}(x)$ over $x$ from the support of $\rho_0$. The first term is small when $\psi_1$ and $\psi_0$ are close to each other, so it remains to study the ratio $\mathcal T_T (\psi_1(x) - \psi_0(x)) e^{\psi_s(x)} / \mathcal T_T e^{\psi_s(x)}$. Due to the Cauchy-Schwarz inequality, it holds that $\mathcal T_T (\psi_1(x) - \psi_0(x)) e^{\psi_s(x)} / \mathcal T_T e^{\psi_s(x)} \leq \sqrt{\mathcal T_T (\psi_1(x) - \psi_0(x))^2} \cdot \sqrt{\mathcal T_T e^{2\psi_s(x)}} /  \mathcal T_T e^{\psi_s(x)}$.
> > > Since $\psi_s = s \psi_1 + (1 - s) \psi_0$ is a $L$-Lipschitz function on $\mathbb R^d$, Herbst's argument and the log-Sobolev inequality (for the Gaussian measure behind the OU operator $\mathcal T_T$) imply that $0.5 \log \mathcal T_T e^{2\psi_s(x)} - \mathcal T_T e^{\psi_s(x)} \lesssim L^2$. This yields that $F'(s) \lesssim abs(\psi_1(y) - \psi_0(y)) + \sup_{x \in supp(\rho_0)} \sqrt{\mathcal T_T (\psi_1(x) - \psi_0(x))^2}$, where $abs(\cdot)$ stands for the absolute value. Then $abs(\log \rho_T^{\psi_1}(y) - \log \rho_T^{\psi_0}(y)) \lesssim abs(\psi_1(y) - \psi_0(y)) + \sup_{x \in supp(\rho_0)} \sqrt{\mathcal T_T (\psi_1(x) - \psi_0(x))^2}$. This will be enough to derive the uniform concentration inequality at the end of Step 6 without assumptions on $T$.
> > >
> > > However, we would like to note that the requirement that $\psi$ is Lipschitz on the whole space $\mathbb R^d$ seems to be very severe. For instance, it is not satisfied when both initial and target distributions are Gaussian. In this case, we have that $\psi$ is a quadratic function, which is not Lipschitz on $\mathbb R^d$. For this reason, we use milder Assumptions 3 and 4 in our work. The large $T$ condition arises as a consequence of a more general setup.

---

> > > ### Comment · Reviewer_LmKh · 2025-08-07
> > >
> > > Dear authors,
> > >
> > > Thank you for these clarifications. I am happy of the answer. I just would like to point out that some of the regularity estimates for potentials are valid also in the setting where one marginal is Gaussian and the other merely of bounded support, see e.g. Prop 1.6 in Chiarini et al. You may find these estimates useful.

---

> ### Comment · Reviewer_LmKh · 2025-08-07
>
> Thank you for clarifying this point. I believe it is worthwhile to stress in the paper that the large $T$ assumption can be circumvented if potentials are regular. Lipschitz may be a lot to ask but if the target is compactly supported, one may expect this to hold. The same comment holds for second order estimates. If the target is compactly supported one can at least expect semi concavity, given that the source is Gaussian according to the reference I pointed out. Please comment on this in the final version of your work

---

### Official Review · Reviewer_CeuM · 2025-07-01

**Clarity:** 3
**Significance:** 3
**Originality:** 3
**Rating:** 5
**Confidence:** 3

**Summary:**

The authors study the problem of estimating the terminal potential in the Schrodinger bridge problem from samples. For a class of multivariate Ornstein-Uhlenbeck reference dynamics, the authors derive an upper bound on the error of the estimated potential within a parametric class of potentials.

**Questions:**

Lines 305-306: “we focus on the statistical error leaving study of the approximation out of the of the present paper.” This is understandable, but I am wondering whether anything can be said at all (e.g. is the GMM family proposed by Korotin et al. rich enough to approximate arbitrarily well a smooth potential?) If this is the case, it would be useful to point this out for non-expert readers.

Do the existing results (namely, Theorem 1) apply to more general $p_0$? While Gaussian $p_0$ is indeed the relevant one for generative modelling, there are numerous applications (e.g. physical sciences, unpaired translation) where both the initial and terminal densities are general.

The discussion of the SBP is very general, allowing for even a diffusivity $\sigma$ that depends on $(t, x)$. However this is used nowhere else in the article. I don’t have a strong opinion on this, but the authors may consider simplifying to the constant case to lighten the notations since they do not end up playing a role in the article itself.

While the current contributions of the paper are no doubt relevant to the practical setting of estimating Schrodinger potentials, it would be valuable to have some numerical validation of the statistical rates derived, even if in extremely simplistic settings (Gaussian/1D/etc). However I understand that these rates may be difficult to verify numerically in practice, so this remains a suggestion.

**Ethical Concerns:**

["NO or VERY MINOR ethics concerns only"]

**Final Justification:**

There were several issues that were clarified in the rebuttal, including several statements which needed to be rectified - the authors have committed to fixing these. Importantly, the contribution of handling a Gaussian initial condition with unbounded support has been made clear to me, while I had initially missed this in my original review. I have increased my score as a result.

**Limitations:**

yes

**Quality:**

3

**Strengths And Weaknesses:**

Strengths:

Overall I found the paper well written and addresses an important problem in the generative modelling literature. The introduction provides a good overview of the SBP in a general setting. The study of the case of parametric family of Schrodinger potentials provides a complementary approach to the analysis of the SBP and is relevant in practice. In particular, study of the empirical risk minimizer is well motivated.

Discussion of related work is quite thorough and the comparison to existing results for non-parametric potential estimation puts the present work in context.

Weaknesses:
In some places, I felt like certain statements are susceptible to misinterpretation.  For instance, lines 75-77: “the Sinkhorn approach … may not achieve the best possible solution for the full problem”. It is unclear specifically what this refers to – Sinkhorn-Knopp solves exactly the static SBP when both source and target measure are discrete. On the other hand, in the statistical setting it provides an empirical estimator for the “true” potentials. Please clarify what is meant by “best” here, i.e. statistically optimal in some sense. In my understanding, the interest of the present problem is the parametric estimation problem which is fundamentally different to the problem aimed to be solved by e.g. Sinkhorn.

Similarly, in lines 140-142 the authors claim that the Sinkhorn algorithm requires quadratic costs. This is not true to my knowledge, since the static SBP is linked to the dynamic (i.e. path space) SBP whenever the bridges of the reference process and its transition kernel are tractable.

While the introduction and preliminaries discuss the SBP setting of general initial and terminal densities, the statement of Theorem 1 currently assumes that $p_0$ is standard Gaussian. There seems to be a disconnect between the result and the earlier discussion. If extension to more general $p_0$ is straightforward, this would need to be clearly stated with some discussion of assumptions. As it currently stands, the result seems to be mostly relevant to the generative modelling scenario.

There appears to be a typo in the abstract: line 12 states $O(log n/n)$ although in Thm. 1 the dominating term should be $\log^2(n) / n$.

---

> ### Author Rebuttal · Authors · 2025-07-30
>
> ### 1. In some places, I felt like certain statements are susceptible to misinterpretation. For instance, lines 75-77: “the Sinkhorn approach … may not achieve the best possible solution for the full problem”. It is unclear specifically what this refers to – Sinkhorn-Knopp solves exactly the static SBP when both source and target measure are discrete. On the other hand, in the statistical setting it provides an empirical estimator for the “true” potentials. Please clarify what is meant by “best” here, i.e. statistically optimal in some sense. In my understanding, the interest of the present problem is the parametric estimation problem which is fundamentally different to the problem aimed to be solved by e.g. Sinkhorn.
>
> We thank the reviewer for pointing out the potential ambiguity in our statement on lines 75–77. Our intention was not to question the correctness of the Sinkhorn algorithm in the fully discrete setting—indeed, as the reviewer rightly notes, Sinkhorn-Knopp solves the static Schrödinger bridge problem exactly when both marginals are discrete. Rather, our comment was meant to highlight a gap in the statistical understanding of the method when applied to empirical data. In particular, while Sinkhorn applied to empirical measures does yield an estimator of the Schrödinger potentials, we are not aware of results showing that this procedure yields nonparametric estimators of the true continuous potentials with provable statistical convergence rates of order $1/n$ (up to logarithmic factors), when the estimation error is measured in terms of Kullback-Leibler (KL) divergence between marginal endpoint distributions.  This rate is minimax optimal for statistical error in KL, and our Theorem 1 establishes that it is achievable in our setting (in the absence of approximation error). If such results exist for Sinkhorn-based estimators, we would be very interested to learn about them and would appreciate any pointers the reviewer could provide. We are aware of recent theoretical advances on the analysis of the Sinkhorn algorithm in more general settings beyond the classical discrete case (see, for example, [Conforti et al., 2023]). These results provide important insights into the convergence behavior of the algorithm in continuous settings with more general reference processes. However, they do not address statistical error or estimation from finite data.
>
> ### 2. Similarly, in lines 140-142 the authors claim that the Sinkhorn algorithm requires quadratic costs. This is not true to my knowledge, since the static SBP is linked to the dynamic (i.e. path space) SBP whenever the bridges of the reference process and its transition kernel are tractable.
>
> We thank the reviewer for this clarification. It is true that the Sinkhorn algorithm, or more generally iterative scaling methods, can be extended beyond the quadratic cost setting and applied to static Schrödinger bridge problems linked to general reference processes—provided the bridges and transition kernels are tractable. We fully agree that the formulation itself is not limited to the quadratic case. Our intention in lines 140–142 was to emphasize that, while these generalizations exist, most of the available statistical analysis-in particular, results concerning convergence guarantees, contraction rates, or finite-sample behavior-has been developed in the setting where the cost is quadratic, corresponding to a Gaussian reference process. To the best of our knowledge, rigorous statistical guarantees (such as nonparametric convergence rates for potentials or couplings estimated from data) have not yet been established in the general case beyond the quadratic cost. This is in contrast to our approach, which provides such guarantees under mild assumptions on marginal distribution and OU reference process. We will revise the text to clarify this point and avoid the misleading impression that the Sinkhorn algorithm cannot be used with more general reference processes.
>
> ### 3. While the introduction and preliminaries discuss the SBP setting of general initial and terminal densities, the statement of Theorem 1 currently assumes that $p_0$ is standard Gaussian. There seems to be a disconnect between the result and the earlier discussion. If extension to more general $p_0$ is straightforward, this would need to be clearly stated with some discussion of assumptions. As it currently stands, the result seems to be mostly relevant to the generative modelling scenario.
>
> In the present paper, we focus on the problem of generative modelling, where, to the best of our knowledge, the Gaussian noise is a standard choice for the initial distribution. However, we can prove a counterpart of Lemma B.2 for any $\rho_0$, which is bounded and has a compact support. We would like to note that initial Gaussian distribution does not simplify but complicates the proof, because it has an unbounded support. In particular, the proof of Lemma B.2 would be much simpler if we assumed that the initial density $\rho_0$ has a compact support. We will add the corresponding discussion after revision.
>
> ### 4. There appears to be a typo in the abstract: line 12 states $O(\log n/n)$ although in Thm. 1 the dominating term should be $\log^2(n) / n$.
>
> Thank you for pointing out at this typo. We will fix it in the revised version.
>
> ### 5. Lines 305-306: “we focus on the statistical error leaving study of the approximation out of the of the present paper.” This is understandable, but I am wondering whether anything can be said at all (e.g. is the GMM family proposed by Korotin et al. rich enough to approximate arbitrarily well a smooth potential?) If this is the case, it would be useful to point this out for non-expert readers.}
>
> Yes, in [Korotin et al., 2024], the authors prove that if the initial and final densities are compactly supported, then for any $\delta > 0$ there exists a Gaussian mixture parametrization that the KL-divergence between the corresponding distribution and the target one does not exceed $\delta$ (see their Theorem 3.4). We will mention this result in the revised version.
>
> ### 6. Do the existing results (namely, Theorem 1) apply to more general $p_0$? While Gaussian $p_0$ is indeed the relevant one for generative modelling, there are numerous applications (e.g. physical sciences, unpaired translation) where both the initial and terminal densities are general.
>
> In the revised version, we will discuss how the result of Theorem 1 can be extended to the case when $\rho_0$ has a compact support. In particular, the proof Lemma B.2 will significantly simplify in this case. In the present paper, we focused on the problem of generative modelling. In unpaired image-to-image translation, one has to use a different target functional minimizing the KL-divergence between couplings (see, for instance, [Korotin et al., ICLR, 2024]). This functional has different properties compared to the KL-divergence between the endpoint densities and, as a consequence, requires different tools for its analysis. In particular, in the unpaired image-to-image translation problem, one cannot rely Lemmata C.1, C.2, and C.3 helping to establish a Bernstein-type condition. Instead, one has to use an inequality relating the variance and the cumulant generating function of a sub-exponential random variable. We will provide all the details in our forthcoming paper.
>
> ### 7. The discussion of the SBP is very general, allowing for even a diffusivity $\sigma$ that depends on $(t, x)$. However this is used nowhere else in the article. I don’t have a strong opinion on this, but the authors may consider simplifying to the constant case to lighten the notations since they do not end up playing a role in the article itself.}
>
> We thank the reviewer for this helpful observation. We agree that allowing for general, time- and space-dependent diffusivity $\sigma(t, x)$ introduces unnecessary complexity in the notation, especially since this generality is not used in the remainder of the paper.
>
> In response to the suggestion, we have simplified the setting to the case of constant diffusivity to improve clarity and streamline the presentation. We believe this change makes the exposition more accessible without affecting the scope of our results.
>
>
> ### 8. While the current contributions of the paper are no doubt relevant to the practical setting of estimating Schrodinger potentials, it would be valuable to have some numerical validation of the statistical rates derived, even if in extremely simplistic settings (Gaussian/1D/etc). However I understand that these rates may be difficult to verify numerically in practice, so this remains a suggestion.}
>
> We thank the reviewer for this thoughtful and constructive suggestion. We fully agree that numerical validation of the statistical convergence rates-even in simplified settings such as one-dimensional or Gaussian marginals-would provide valuable support to the theoretical results and enhance the practical relevance of the paper.
> While verifying the $1/n$ statistical rate empirically can be challenging due to the need to disentangle approximation and optimization errors, we view this as an important direction for future work. In particular, low-dimensional cases offer a feasible benchmark for isolating the statistical component of the error and could help illustrate the theoretical behavior in practice. We appreciate the reviewer’s understanding and will note this suggestion in the conclusion as a natural next step in the development of our framework.
>
> [Conforti et al., 2023] G. Conforti, A. Durmus, G. Greco. Quantitative contraction rates for Sinkhorn algorithm: beyond bounded costs and compact marginals. ArXiv:2304.04451, 2023.
>
> [Korotin et al., 2024] A. Korotin, N. Gushchin, E. Burnaev. Light Schrödinger Bridge. In The Twelfth International Conference on Learning Representations, 2024.

---

> ### Comment · Reviewer_CeuM · 2025-08-03
>
> Thank you for clarifying all these points, these have improved my understanding of the paper and its contributions. I trust that the ambiguities in writing will be improved and I see that the handling of the Gaussian initial condition is technically more involved compared to those assuming compact support. It would be valuable, in my opinion, to make this very clear in the paper, since it was not so clear to me.
> Given the discussion, I understand the contributions of the paper more clearly and will raise my score.

---

### Official Review · Reviewer_j9qF · 2025-07-02

**Clarity:** 3
**Significance:** 2
**Originality:** 3
**Rating:** 4
**Confidence:** 1

**Summary:**

This article deals with the problem of approximating the boundary potentials of the Schrödinger bridge problem through sampling of the marginal distributions. The authors formulate the approximation as an empirical Kullback–Leibler (KL) risk minimization over a suitable class of log-potentials, where the goal is to recover the boundary potentials that couple the marginals through the dynamics of the Schrödinger bridge. The main theoretical contribution is the derivation of a KL convergence rate of $\mathcal{O}\left( \log(n)/n \right)$ with high probability as the sample size $n$ increases, under mild assumptions on the target density, such as boundedness and sub-Gaussian tails.

**Questions:**

(1) It is not clear how to approach the existence and uniqueness of the minimization problem (3). Is it a convex optimization problem over the class of log-potentials?

(2) What kind of numerical methods could be implemented to compute $\hat{\rho}_T$? What is the computational cost for solving the coupled problem (3), and are there any available error estimates?

(3) It seems that the main estimate (lines 277–279) suffers from the curse of dimensionality ($\approx \mathcal{O}(d^2)$), making the problem difficult to scale to higher-dimensional settings. How does this compare with the Sinkhorn algorithm or other existing Schrödinger potential estimators?

**Ethical Concerns:**

["NO or VERY MINOR ethics concerns only"]

**Final Justification:**

While the paper does not include a numerical algorithm, its theoretical contributions are significant and provide meaningful insights into the problem. This justifies maintaining a positive evaluation.

**Limitations:**

Yes.

**Paper Formatting Concerns:**

None.

**Quality:**

2

**Strengths And Weaknesses:**

Strength:
The article is mathematically sophisticated and sound. The assumptions on the target density, namely boundedness and sub-Gaussian tails, are mild and cover a broad range of applications. The formulation couples all elements of the Schrödinger bridge problem, including the marginals, transition densities, and potential function. The approximation is straightforward, aiming to minimize the KL divergence directly, in contrast to popular methods like the Sinkhorn algorithm, which decouple the marginals.

Weakness:
The empirical risk minimization problem appears challenging to solve, especially as the sample size increases. Moreover, the paper does not provide numerical experiments to demonstrate the convergence of the estimator in practice. In addition, the main estimate (equation 278) is not dimension-free, the error is of order $\mathcal{O}(d^2)$, which suggests that the method may be difficult to scale for large data in real applications.

---

> ### Author Rebuttal · Authors · 2025-07-30
>
> ### 1. The empirical risk minimization problem appears challenging to solve, especially as the sample size increases. Moreover, the paper does not provide numerical experiments to demonstrate the convergence of the estimator in practice. In addition, the main estimate (equation 278) is not dimension-free, the error is of order $\mathcal{O}(d^2)$, which suggests that the method may be difficult to scale for large data in real applications.
>
> We thank the reviewer for raising this important practical question. In practice, the integral with respect to the marginal distribution $\rho_0$ can be approximated using Monte Carlo sampling, which is natural when the data consist of independent samples from $\rho_0$. The time-evolved forward Schrödinger potential $h$ is then computed by optimizing over a suitable function class $\Psi$ of log-potentials, for example neural networks or other flexible parametric families.
> That said, we acknowledge that developing efficient and scalable algorithms for this optimization problem-especially in high dimensions or under structural constraints-is a significant challenge. While our current work focuses on the theoretical aspects and statistical guarantees, addressing the computational side of the problem is an important direction for future research, and we plan to explore it in more depth in subsequent work.
>
> We also would like to thank the reviewer for pointing out to quadratic dependence on the dimension. After careful inspection of the proof, we figured out that the bound $\mathcal O(d^2)$ can be  improved to $\mathcal O(d \log d)$. This is due to a more careful analysis eliminating the factor $e^{\mathcal O(d)}$ in Lemma B.2. The dependence on the dimension $\mathcal O(d \log d)$ is quite mild, especially taking into account that we impose very loose assumptions on the target density $\rho_T^\star$.
> To our knowledge, dimension-free bounds usually require additional assumptions, such as $L_2$-$\psi_2$ equivalence in linear regression, which is hard to satisfy in the nonlinear setup. Dimension-dependent rates are also usual for generative diffusion models (see, e.g., [Benton et al., 2024] and [Conforti et al., 2025]). We doubt that one can easily eliminate dependence on the dimension.
>
>
> ### 2. It is not clear how to approach the existence and uniqueness of the minimization problem (3). Is it a convex optimization problem over the class of log-potentials?
>
> To guarantee the existence of the solution of the minimization problem, it is enough to require the class of log-potentials $\Psi$ be closed (in addition to the assumptions we list in our paper). Then the existence follows from the continuity of the objective with respect to the log-potential. We have not studied the questions of uniqueness and convexity. However, we would like to note that if one takes $\Psi$ to be a class of deep neural networks, one inevitably faces a non-convex optimization problem with a very complicated loss landscape having a lot of local minima and, possibly, several global minima. However, this situation is ubiquitous in deep learning.
>
>
> ### 3. What kind of numerical methods could be implemented to compute $\hat{\rho}_T$? What is the computational cost for solving the coupled problem (3), and are there any available error estimates?
>
> We thank the reviewer for raising this important practical question. In practice, the integral with respect to the marginal distribution $\rho_0$ can be approximated using Monte Carlo sampling, which is natural when the data consist of independent samples from $\rho_0$. The time-evolved forward Schrödinger potential $h$ is then computed by optimizing over a suitable function class $\Psi$ of log-potentials, for example neural networks or other flexible parametric families. However, we agree that this question is important and challenging and would like to address it in our future work. The computational cost and error estimates will heavily depend on the class of log-potentials $\Psi$. In particular, we are not sure if one can provide any guarantees on the accuracy of the solution of the optimization problem when $\Psi$ is a class of neural networks.
>
> ### 4. It seems that the main estimate (lines 277–279) suffers from the curse of dimensionality ($\approx \mathcal{O}(d^2)$), making the problem difficult to scale to higher-dimensional settings. How does this compare with the Sinkhorn algorithm or other existing Schr\"odinger potential estimators?
>
> As we mentioned earlier, the dependence $\mathcal O(d^2)$ can be improved to $\mathcal O(d \log d)$. We will incorporate the corresponding changes into the revised version. In [Pooladian, Niles-Weed, 2024], the authors consider the Sinkhorm algorithm and obtain exponential dependence on the dimension when the initial and final distributions are supported on sets with non-empty interiors. In a very relevant paper [Korotin et al., 2024], the authors tracked  dependence on the sample size only and left $d$ in the hidden constants. We also would like to note that the nearly linear dependence $\mathcal O(d \log d)$ can also be found in  the analysis of generative diffusion models (see [Benton et al., 2024] and [Conforti et al., 2025]) where the KL-error scales as $\mathcal O(d)$.
>
> [Benton et al., 2024] J. Benton, V. D. Bortoli, A. Doucet, and G. Deligiannidis. Nearly d-linear convergence bounds for diffusion models via stochastic localization. In The Twelfth International Conference on Learning Representations, 2024.
>
> [Conforti et al., 2025] G. Conforti, A. Durmus, M. G. Silveri. KL Convergence Guarantees for Score Diffusion Models under Minimal Data Assumptions. SIAM Journal on Mathematics of Data Science 7(1), pp. 86-109, 2025.
>
> [Korotin et al., 2024] A. Korotin, N. Gushchin, E. Burnaev. Light Schrödinger Bridge. In The Twelfth International Conference on Learning Representations, 2024.
>
> [Pooladian, Niles-Weed, 2024] A.-A. Pooladian, Niles-Weed. Plug-in estimation of Schrödinger bridges. ArXiv:2408.11686, 2024.

---

> > ### Comment · Reviewer_j9qF · 2025-08-06
> >
> > Thank you for the detailed clarifications. The author has provided reasonable explanations and discussed challenges related to my questions. Although the paper lacks a numerical algorithm, its theoretical contributions are valuable. Accordingly, I will raise my score.

---

### Official Review · Reviewer_MJbQ · 2025-07-03

**Clarity:** 3
**Significance:** 3
**Originality:** 3
**Rating:** 5
**Confidence:** 2

**Summary:**

The paper derives non-asymptotic sample-complexity bounds showing that empirical KL-risk minimization can learn the Schrödinger potential governing a multivariate Ornstein–Uhlenbeck bridge under only sub-Gaussian data assumptions. It proves the potential converges at a rate that scales like the squared logarithm of the sample size divided by the sample size, improving on earlier Sinkhorn-based guarantees and reinforcing the practicality of typical case of Schrödinger-bridge generative models.

**Questions:**

* The paper assumes a standard Gaussian for the initial distribution, yet real-world datasets rarely follow that law. This mismatch is precisely why Schrödinger-bridge techniques appeal to machine-learning practitioners. The Gaussian start certainly simplifies the mathematics, but the manuscript does not spell out the concrete difficulties that arise when the starting law is non-Gaussian. Clarifying those obstacles and outlining how the proofs would need to adapt once the Gaussian assumption is dropped would considerably strengthen the contribution.

* I am familiar with the Schrödinger‐bridge framework and have used various type of solvers like IPF and IMF, yet I’m unsure how Equation (3) is meant to be applied in practice. Practitioners typically favor Sinkhorn‐style or IMF objectives because they are easier to implement, so I may be missing something. Could you clarify whether Equation (3) can realistically function as a training objective?

**Ethical Concerns:**

["NO or VERY MINOR ethics concerns only"]

**Final Justification:**

The paper delivers rigorous, non-asymptotic sample-complexity bounds showing that empirical KL-risk minimization can learn the Schrödinger potential under merely sub-Gaussian data assumptions. The convergence rate $\tilde O(\log^{2} n / n)$ improves on earlier Sinkhorn-based guarantees and reinforces the practicality of Schrödinger-bridge generative models in the typical Gaussian-noise setting. Although Equation (3) is not yet directly practical for large-scale training, the theoretical framework established here is a clear milestone toward future scalable solvers for non-Gaussian settings. Given these contributions and clarifications, I find the strengths outweigh the remaining concerns and therefore I keep my score.

**Limitations:**

yes

**Quality:**

3

**Strengths And Weaknesses:**

**Strengths**

* The paper is clearly structured and easy to follow. Each lemma appears with a concise explanation of its implications.

* The authors replace the alternating Sinkhorn updates in favor of a single empirical-risk minimization, learning the forward and backward Schrödinger potentials in one coherent step.

* Theorem 1 shows, under remarkably weak regularity assumptions, that the excess KL risk of the empirical minimizer decays quickly, delivering faster convergence than earlier analyses.

----

**Weakness**

See questions.

---

> ### Author Rebuttal · Authors · 2025-07-30
>
> ### 1. The paper assumes a standard Gaussian for the initial distribution, yet real-world datasets rarely follow that law. This mismatch is precisely why Schrödinger-bridge techniques appeal to machine-learning practitioners. The Gaussian start certainly simplifies the mathematics, but the manuscript does not spell out the concrete difficulties that arise when the starting law is non-Gaussian. Clarifying those obstacles and outlining how the proofs would need to adapt once the Gaussian assumption is dropped would considerably strengthen the contribution.
>
> In the present paper, we focus on the problem of generative modelling, where, to the best of our knowledge, the Gaussian noise is a standard choice for the initial distribution. However, we can prove a counterpart of Lemma B.2 for any $\rho_0$, which is bounded and has a compact support. We would like to note that initial Gaussian distribution does not simplify but complicates the proof, because it has an unbounded support. In particular, the proof of Lemma B.2 would be much simpler if we assumed that the initial density $\rho_0$ has a compact support. We will add the corresponding discussion after revision.
>
> We also would like to mention that we do not consider the problem of unpaired image-to-image translation, where the initial distribution is non-Gaussian (and, in fact, is unknown). In this case, one should minimize the KL-divergence between couplings (see, for instance, [Korotin et al., ICLR, 2024]). This functional has different properties compared to the KL-divergence between final distributions and requires a different analysis. We will provide it in our forthcoming work.
>
>
>
>
> ### 2. I am familiar with the Schrödinger‐bridge framework and have used various type of solvers like IPF and IMF, yet I’m unsure how Equation (3) is meant to be applied in practice. Practitioners typically favor Sinkhorn‐style or IMF objectives because they are easier to implement, so I may be missing something. Could you clarify whether Equation (3) can realistically function as a training objective?
>
>
> We thank the reviewer for sharing their perspective. It is true that Equation (3) may appear more complex compared to Sinkhorn-style or iterative proportional fitting (IPF/IMF) formulations commonly used in practice.
> However, it’s important to note that many of these practical algorithms—including Sinkhorn, IPF, and IMF—are specifically tailored to the case where the reference process is Gaussian, corresponding to a quadratic cost in the static formulation. In such settings, the Schrödinger system simplifies substantially and admits efficient solvers exploiting the structure of the cost and marginals. In contrast, our framework is designed to handle more general reference processes, beyond Brownian motion. Equation (3) reflects the KL divergence from the learned terminal distribution to the desired one, and is thus naturally suited to cases where the underlying diffusion has more complex dynamics or reflects domain-specific prior knowledge. While this generality comes at a computational cost, it allows for greater modeling flexibility. We agree that more work is needed to make these general formulations practical, and we view the development of scalable solvers for non-Gaussian reference processes as an important  direction for future research.
>
> [Korotin et al., 2024] A. Korotin, N. Gushchin, E. Burnaev. Light Schrödinger Bridge. In The Twelfth International Conference on Learning Representations, 2024.

---

> ### Comment · Reviewer_MJbQ · 2025-08-04
>
> Thank you for the detailed clarifications. Although I remain unconvinced of its immediate practicality, the theoretical foundations laid here mark an important milestone toward practical algorithm for a general formulation that the authors stated. Accordingly, I will raise my score.

---

### Official Review · Reviewer_Lo7j · 2025-07-07

**Clarity:** 3
**Significance:** 2
**Originality:** 2
**Rating:** 4
**Confidence:** 3

**Summary:**

This paper provides a theoretical analysis of the sample complexity for estimating the Schrödinger potential in the context of Schrödinger bridges. The authors study an estimator based on empirical KL risk minimization, using a multivariate Ornstein-Uhlenbeck (OU) process as the reference. The main contribution is a non-asymptotic, high-probability upper bound on the excess KL risk of the estimated terminal distribution. This bound demonstrates that a "fast" convergence rate, potentially as rapid as $O(\log^2 n/n)$ can be achieved. This is a significant improvement over standard rates and is established under relatively weak assumptions, such as allowing for unbounded supports for the initial and terminal distributions.

**Questions:**

- How does the sample complexity rate degrade as $bT$ approaches zero? Does the analysis break down, or does it gracefully transition to a slower rate? Clarifying this would significantly broaden the applicability of your results.
- Can the analysis be refined to replace the explicit dependence on the parameter count $D$ with a more nuanced complexity measure, thereby making the bound meaningful for over-parameterized models?
- What is the trade-off between the structural assumptions on your function class $\Psi$ (required for the theory) and its ability to achieve a small approximation error for complex targets? Furthermore, how can the non-trivial optimization problem be addressed in practice?

**Ethical Concerns:**

["NO or VERY MINOR ethics concerns only"]

**Final Justification:**

My concerns have been resolved, and I am now slightly positive about acceptance. The only reason my score is not higher is due to my assessment of the significance of the theoretical contribution.

**Limitations:**

yes

**Quality:**

3

**Strengths And Weaknesses:**

Strengths:
- The paper delivers a significant theoretical result (a fast-rate convergence guarantee) for a fundamental problem in generative modeling. The technical proofs are of high quality, successfully handling challenging, realistic settings.
- The paper is well-written, with clear assumptions, results, and a cogent comparison to related work.
Weakness:
- The theory requires $bT$ to be large, limiting its applicability to "long" bridge problems and excluding short-time or slow-dynamics scenarios.
- The paper completely ignores the significant computational challenge of optimizing the complex, integral-based objective function. I doubt that there is an algorithm that can achieve this. Based on the results of the existing IMF and IPF, I think it is pretty challenging.

---

> ### Author Rebuttal · Authors · 2025-07-30
>
> ### 1. The theory requires $bT$ to be large, limiting its applicability to "long" bridge problems and excluding short-time or slow-dynamics scenarios. How does the sample complexity rate degrade as $bT$ approaches zero? Does the analysis break down, or does it gracefully transition to a slower rate? Clarifying this would significantly broaden the applicability of your results.
>
> If $bT$ tends to zero, the analysis does not break down completely. In particular, Lemma C.1 and C.2 establishing the Bernstein-type condition for the log-density loss do not require any conditions on $bT$ and remain valid when $bT$ approaches zero. The assumption on $bT$ plays a crucial role in  Lemma B.2, which ensures that the difference $\log \rho_T^\psi - \log \rho_T^\varphi$ is small whenever the corresponding log-potentials $\psi$ and $\varphi$ are close to each other. If $bT$ tends to zero, one must find a different way to bound $\log \rho_T^\psi - \log \rho_T^\varphi$. We will discuss it in the revised version. However, we would like to note that we only require $T = \Omega(1)$ while in analysis of score-based diffusion models (see, e.g., [Oko et al., 2023], [Benton et al., 2024], [Conforti et al., 2025]), one has to take $T$ of order $\log n$ to ensure that the error tends to zero polynomially as the sample size $n$ grows. In this sense, ''long'' bridge problems remain much ``shorter'' than ordinary score-based diffusion models.
>
> ### 2. The paper completely ignores the significant computational challenge of optimizing the complex, integral-based objective function. I doubt that there is an algorithm that can achieve this. Based on the results of the existing IMF and IPF, I think it is pretty challenging.
>
> We appreciate the reviewer's comment regarding the computational complexity of our approach. Indeed, the optimization problem we consider involves integral-based terms and is inherently nontrivial. However, this complexity is not unique to our method-it is intrinsic to the Schrödinger problem itself. The core difficulty lies in the nonlinearity of the Schrödinger system, which any algorithm aiming to estimate the potentials must address in some form. For instance, the widely used Sinkhorn algorithm also faces this challenge: it attempts to solve the underlying fixed-point equations (arising from the Schrödinger system with some reference transition kernel) via Picard iterations. While Sinkhorn is efficient in some settings, its convergence depends heavily on contraction properties in Hilbert’s metric  and can become quite slow or unstable when integrals in each iteration are approximated-particularly in continuous or high-dimensional spaces. Our approach trades direct fixed-point iteration for variational optimization, which, while computationally demanding, allows us to integrate statistical learning techniques (e.g., stochastic optimization and function approximation via neural networks) and work with samples rather than discretized grids. This opens the door to handling high-dimensional problems in a flexible, data-driven manner for general reference processes. Also note that the main goal of the present submission was to establish sharper theoretical bounds on the accuracy of Schr\"odinger potential estimation. We would like to address computational issues in the future work.
>
> ### 3. Can the analysis be refined to replace the explicit dependence on the parameter count $D$ with a more nuanced complexity measure, thereby making the bound meaningful for over-parameterized models?
>
> We suppose that one may obtain similar bounds using the metric entropy of the class of log-potentials. However, at the present moment, we cannot obtain meaningful bounds for the over-parametrized regime, because of lack of machinery. To our knowledge, theoretical foundations of over-parametrized models mostly concern the case of linear regression or regression with shallow neural networks and do not consider deep neural networks.
>
> ### 4. What is the trade-off between the structural assumptions on your function class $\Psi$ (required for the theory) and its ability to achieve a small approximation error for complex targets? Furthermore, how can the non-trivial optimization problem be addressed in practice?
>
> We thank the reviewer for raising these important questions. The approximation-estimation trade-off depends on properties of the true log-potential $\psi^\star$. In the end of Section 4, we note that it is enough to approximate the true log-potential $\psi^\star$ with respect to $L_1(\rho_T^\star)$- and $L_1(\mathcal N(m, \Sigma / (2b)))$-distance to achieve a small approximation error with respect to the KL-loss. In order to do so, we need a prior knowledge on smoothness of $\psi^\star(y)$ and its behaviour when $y \rightarrow \infty$. For instance, assume that for any $R> 0$ the function $\psi^\star(y)$ is Lipschitz on $[-R, R]^d$ with a constant $L(R)$ and than $|\psi^\star(y)| \leq A (\|y\|^\alpha + 1)$ for all $y \in \mathbb R^d$. Then we can take a sufficiently large $R$ (of order $\sqrt{\log(1/\varepsilon)}$), approximate $\psi^\star(y)$ on the cube $[-R, R]^d$ using standard results on properties (for instance, [Yarotsky, 2017]) and then bound $L_1(\rho_T^\star)$- and $L_1(\mathcal N(m, \Sigma / (2b)))$-errors using the fact that $\psi^\star(y)$ grows polynomially as $y$ tends to infinity. This means that $L_1$-errors outside the cube $[-R, R]^d$ will tend to zero as $R$ approaches infinity. However, we are not aware of results on smoothness of $\psi^\star$ of this kind in the literature.
>
> In practice, the integral with respect to the marginal distribution $\rho_0$ can be approximated using Monte Carlo sampling, which is natural when the data consist of independent samples from $\rho_0$. The time-evolved forward Schrödinger potential $h$ is then computed by optimizing over a suitable function class $\Psi$ of log-potentials, for example neural networks or other flexible parametric families.
> That said, we acknowledge that developing efficient and scalable algorithms for this optimization problem-especially in high dimensions or under structural constraints-is a significant challenge. While our current work focuses on the theoretical aspects and statistical guarantees, addressing the computational side of the problem is an important direction for future research, and we plan to explore it in more depth in subsequent work.
>
> [Benton et al., 2024] J. Benton, V. D. Bortoli, A. Doucet, and G. Deligiannidis. Nearly d-linear convergence bounds for diffusion models via stochastic localization. In The Twelfth International Conference on Learning Representations, 2024.
>
> [Conforti et al., 2025] G. Conforti, A. Durmus, M. G. Silveri. KL Convergence Guarantees for Score Diffusion Models under Minimal Data Assumptions. SIAM Journal on Mathematics of Data Science 7(1), pp. 86-109, 2025.
>
> [Oko et al., 2023] K. Oko, S. Akiyama, and T. Suzuki. Diffusion models are minimax optimal distribution estimators. In
> International Conference on Machine Learning, pp. 26517–26582. PMLR, 2023.
>
> [Yarotsky, 2017] D. Yarotsky. Error bounds for approximations with deep ReLU networks. Neural networks 94, 103-114.

---

> > ### Comment · Reviewer_Lo7j · 2025-08-06
> > **Response to the rebuttal**
> >
> > Thank you for the constructive rebuttal. Your answers have satisfactorily addressed my questions, and I will maintain my current score.

---

### Author Response · Authors · 2025-08-08

We are grateful to the reviewers for the constructive feedback. In the revised version, we will address the issue with quadratic dependence on $d$, elaborate on further discussion about Sinkhorn’s algorithm, and add the references suggested by the reviewers. We will also show that the large $T$ requirement can be relaxed and provide insights into the quantification of the approximation error in the case of Lipschitz log-potentials. We believe that the paper will benefit from these changes.

---

### Decision · Program_Chairs · 2025-09-17

**Decision:**

Reject

**Comment:**

This paper studies the problem of Schrödinger potential estimation, which plays a central role in generative modeling methods based on Schrödinger bridges and stochastic optimal control. All reviewers agree that this is a strong paper and recommend acceptance.

The reviewers all found that the merits of the work are clear. It establishes rigorous fast-rate convergence guarantees for Schrödinger potential estimation under sub-Gaussian assumptions, improving on earlier Sinkhorn-based results and demonstrating the feasibility of learning Schrödinger potentials in realistic Gaussian-noise settings. Reviewers highlighted the clarity of the exposition, the precise formulation of assumptions and results, and the careful positioning of the work within related literature. While not directly providing a numerical algorithm, the theoretical framework laid out here represents a milestone for future scalable solvers in more general settings.

Some weaknesses were noted. The theory requires large time horizons, limiting its immediate applicability to long bridge problems. The paper does not address the computational challenges of optimizing the integral-based objective. The comparison with Sinkhorn’s algorithm was also considered imprecise, and reviewers suggested either refining the comparison or providing insights into approximation error. Finally, some issues in phrasing and interpretation were identified during review; these were clarified in the rebuttal, and the authors have committed to fixing them in the final version. Incorporating these improvements will further strengthen the paper.

In sum, this is a clear and rigorous work that advances our theoretical understanding of Schrödinger-bridge generative modeling. With its strong contributions and careful exposition, it represents a valuable addition to the field, and I recommend acceptance.

===

As recently advised by legal counsel, the NeurIPS Foundation is unable to provide services, including the publication of academic articles, involving the technology sector of the Russian Federation’s economy under a sanction order laid out in Executive Order (E.O.) 14024.

Based upon a manual review of institutions, one or more of the authors listed on this paper submission has ties to organizations listed in E.O. 14024. As a result this paper has been identified as falling under this requirement and therefore must not be accepted under E.O. 14024.

This decision may be revisited if all authors on this paper can provide proof that their institutions are not listed under E.O. 14024 to the NeurIPS PC and legal teams before October 2, 2025. Final decisions will be communicated soon after October 2nd. Appeals may be directed to pc2025@neurips.cc.